# Indian Ocean glacial deoxygenation and respired carbon accumulation during mid-late Quaternary ice ages

Liao Chang [1,2] ✉, Babette A. A. Hoogakker [3], David Heslop [4], Xiang Zhao [4], Andrew P. Roberts [4], Patrick De Deckker [4], Pengfei Xue[1], Zhaowen Pei [1], Fan Zeng[1], Rong Huang [1], Baoqi Huang[1], Shishun Wang [1], Thomas A. Berndt[5], Melanie Leng[6,7], Jan-Berend W. Stuut[8] & Richard J. Harrison[9]

Reconstructions of ocean oxygenation are critical for understanding the role of respired carbon storage in regulating atmospheric $CO_2$. Independent sediment redox proxies are essential to assess such reconstructions. Here, we present a long magnetofossil record from the eastern Indian Ocean in which we observe coeval magnetic hardening and enrichment of larger, more elongated, and less oxidized magnetofossils during glacials compared to interglacials over the last ~900 ka. Our multi-proxy records of redox-sensitive magnetofossils, trace element concentrations, and benthic foraminiferal $\Delta\delta^{13}C$ consistently suggest a recurrence of lower $O_2$ in the glacial Indian Ocean over the last 21 marine isotope stages, as has been reported for the Atlantic and Pacific across the last glaciation. Consistent multi-proxy documentation of this repeated oxygen decline strongly supports the hypothesis that increased Indian Ocean glacial carbon storage played a significant role in atmospheric $CO_2$ cycling and climate change over recent glacial/interglacial timescales.

Ocean oxygenation plays an important role in regulating the global carbon budget and major nutrient cycles because of the stoichiometric relationship between oceanic oxygen removal through bacterial respiration (decay of organic matter produced by photosynthesis) and accumulation of this respired organic carbon[1]. Oxygen utilization reflects the strength of the oceanic biological carbon pump that regulates atmospheric $CO_2$ contents[2–4]. Therefore, reconstructions of past ocean oxygenation concentrations provide an important approach to characterize the dynamics of the atmosphere-ocean partitioning of carbon and ocean carbon storage. Boyle[5,6] originally proposed the "nutrient deepening" hypothesis by combining evidence from organic carbon flux and sediment redox proxies and concluded that ocean

oxygenation must have been lower during the last glacial period[7]. Relevant ocean processes were described to explain the lower atmospheric $CO_2$ concentrations during glacials recorded in ice cores[2]. A generally increased respired carbon reservoir in the Pacific and Atlantic Oceans during recent glacials was suggested from subsequent ocean oxygenation reconstructions using organic carbon flux and sediment redox proxies[7–9], sediment magnetic properties[10], benthic foraminiferal carbon isotope gradient[11,12], biogenic organic compounds[13], and more recently reinterpretations of sedimentary manganese concentration[14]. Independent temporal and spatial proxies of ocean oxygenation, particularly long records from other oceans, are crucial to verify this hypothesis.

[1]Laboratory of Orogenic Belts and Crustal Evolution, School of Earth and Space Sciences, Peking University, 100871 Beijing, China. [2]Laboratory for Marine Geology, Qingdao National Laboratory for Marine Science and Technology, 266071 Qingdao, China. [3]The Lyell Centre, Heriot-Watt University, Edinburgh EH14 4BA, UK. [4]Research School of Earth Sciences, The Australian National University, Canberra ACT 2601, Australia. [5]Department of Geophysics, School of Earth and Space Sciences, Peking University, 100871 Beijing, China. [6]National Environmental Isotope Facility, British Geological Survey, Keyworth NG12 5GG, UK. [7]School of Biosciences, University of Nottingham, Sutton Bonington LE12 5RD, UK. [8]NIOZ-Royal Netherlands Institute for Sea Research and Utrecht University, Texel, The Netherlands. [9]Department of Earth Sciences, University of Cambridge, Cambridge CB2 3EQ, UK. ✉e-mail: liao.chang@pku.edu.cn

Several marine sediment proxies have been used to trace past bottom-water oxygenation (BWO) variations. These proxies include sedimentological features (i.e., sediment lamination due to a lack of bioturbation), paleontological data (i.e., morphologies and species abundance of benthic foraminiferal assemblages), the calibrated $\delta^{13}C$ gradient between coeval infaunal and epifaunal benthic foraminifera ($\Delta\delta^{13}C$), redox-sensitive trace-metal concentrations, I/Ca ratio in epifaunal benthic foraminifera, biomarker preservation, and others[7–16] (Supplementary Text S1). An emerging method involves the analysis of magnetofossils−biogenic magnetite nanoparticles that are fossilized after magnetotactic bacteria (MTB) die[17]. MTB mostly live near the water/sediment interface, i.e., typically only a few centimeters below the water/sediment interface in an open ocean setting[18]. Magnetite biomineralization in MTB is genetically controlled but is also redox-sensitive[17–19]. Their fossilized signatures through sediment sequences provide a tracer of sediment/pore water redox conditions, as well as nutrient supply and organic matter cycling[20–26].

In this work, we present a magnetofossil record to reconstruct deep-sea oxygenation variations in the eastern Indian Ocean over the past 21 marine isotope stages (MIS), using a mid-late Quaternary deep-sea sediment core from offshore Western Australia (Fig. 1a). We analyze magnetofossil crystal morphologies using transmission electron microscope (TEM) observations and decomposition of magnetic coercivity to quantify the magnetofossil signals that reflect redox changes of the topmost sediments. The magnetofossil signatures are ultimately linked to BWO variations through oxygen diffusion and indirectly to sediment oxygenation environments through organic matter decomposition. The magnetofossil reconstructions of deep-sea oxygenation are combined with results of quantitative bottom-water oxygen reconstructions using the benthic foraminiferal $\Delta\delta^{13}C$[11,12] and redox-sensitive trace-metal concentration measurements[8,15] on typical glacial/interglacial samples from the same core.

## Results

### Magnetic properties and coercivity distributions

Hysteresis loops and isothermal remanent magnetization (IRM) acquisition curves have contrasting properties for interglacial and glacial samples; glacial samples have relatively wider loops and harder IRM curves. First-order reversal curve (FORC) diagrams[27] for typical samples (Fig. 2a, b) have a similar major central ridge along $B_u = 0$, and a weaker vertical distribution. This FORC signature is typical of a mixture of single domain (SD) biogenic magnetite in chains, collapsed chains, and small amounts of detrital magnetic minerals[25,28,29]. Some large prismatic magnetofossils may behave as vortex state particles

and contribute more to the non-central ridge components of FORC diagrams[29]. Despite the similarity, the central ridge component of glacial samples has a much higher peak coercivity with an extended distribution to higher fields (Fig. 2b; Supplementary Text S2, Fig. S1f−i) compared to those of interglacial samples (Fig. 2a; Supplementary Fig. S1a−e). Such contrasting magnetic behavior between glacial and interglacial samples is observed consistently for samples from core depths above 17.55 m.

Alternating field (AF) demagnetization results for IRM (Fig. 2c, d; Supplementary Text S2, Fig. S2) and anhysteretic remanent magnetization (ARM) data (Supplementary Text S2, Fig. S3) measured on a cryogenic magnetometer indicate well-resolved coercivity components; glacial samples contain a much larger high-coercivity peak than interglacial samples. IRM demagnetization data for typical samples can be fitted well with three components (Fig. 2c, d; Supplementary Fig. S2). The smallest-coercivity component is likely due to a combination of coarse detrital grains, collapsed magnetofossil chains, and magnetostatic interactions[30]. The other two main components (at ~25 and ~65 mT) correspond to the biogenic soft (BS) and biogenic hard (BH) components[31]. A clear BH enrichment is evident in glacial (Fig. 2d; Supplementary Fig. S2f−j) compared to interglacial samples (Fig. 2c; Supplementary Fig. S2a−e).

Low-temperature magnetic measurements of typical samples from interglacial (Fig. 2e) and glacial stages (Fig. 2f) contain a ~100 K Verwey transition ($T_V$) due to magnetofossils[32]. This magnetofossil feature is more prominent for the glacial sample than the interglacial sample. Moreover, divergence between the zero-field-cooled (ZFC) and field-cooled (FC) curves below room temperature due to magnetite oxidation[33], is larger for the interglacial sample (Fig. 2e) than the glacial sample (Fig. 2f). The low-temperature magnetic data consistently suggest that magnetofossils in the interglacial sample are more oxidized than the glacial sample, which is likely related to the more oxic conditions for magnetite biogenesis during interglacials.

### TEM observations and magnetofossil counts

TEM observations on two samples, selected from interglacial (1.25 m core depth; MIS 1) and glacial (3.15 m core depth; MIS 4) stages indicate variable magnetofossil morphologies[17,20–26] (Fig. 3a, e), which are divided into three groups: (1) more equant octahedral and cubo-octahedral crystals, (2) elongated prismatic crystals, and (3) bullet-shaped crystals (color arrows in Fig. 3a, e). Measurements of large sets of magnetofossil crystals indicate a clear distinction between magnetofossil populations for the two glacial and interglacial samples (Fig. 3b, f). Histograms of magnetofossil length and elongation

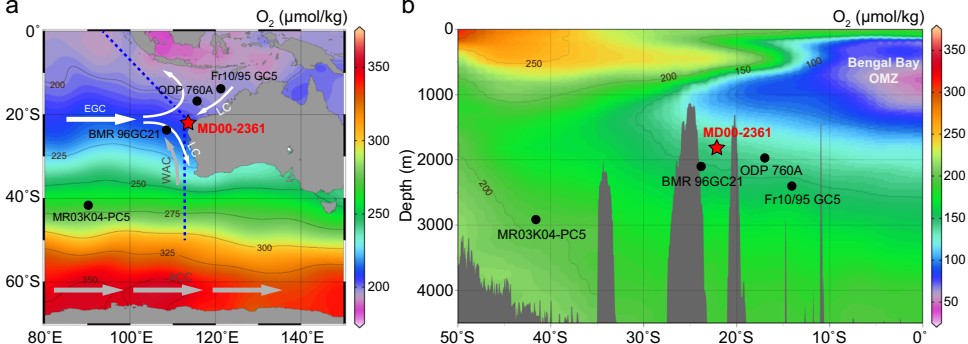

**Fig. 1 | Core location and oceanographic setting. a** Spatial distribution of modern surface-water oxygen concentrations in the southeast Indian Ocean. **b** Hydrographic section of the Indian Ocean across the blue dashed line in **a** with dissolved oxygen levels in the water column. Locations of the studied sediment core MD00-2361 (red star) and comparison cores BMR 96GC21, ODP 760 A, Fr10/95 GC5, and MR03K04-PC5 (black circles) are indicated. Maps are generated in Ocean

Data View (Schlitzer, Reiner, Ocean Data View, https://odv.awi.de; ref. 70), with seawater oxygenation data from the World Ocean Atlas 2013 (ref. 71). Numbers along the isograms indicate $O_2$ concentrations. White and gray arrows indicate surface and deep-water mass flow paths, respectively. LC Leeuwin Current, WAC West Australian Current, EGC East Gyral Current, ACC Antarctic Circumpolar Current, OMZ oxygen minimum zone.

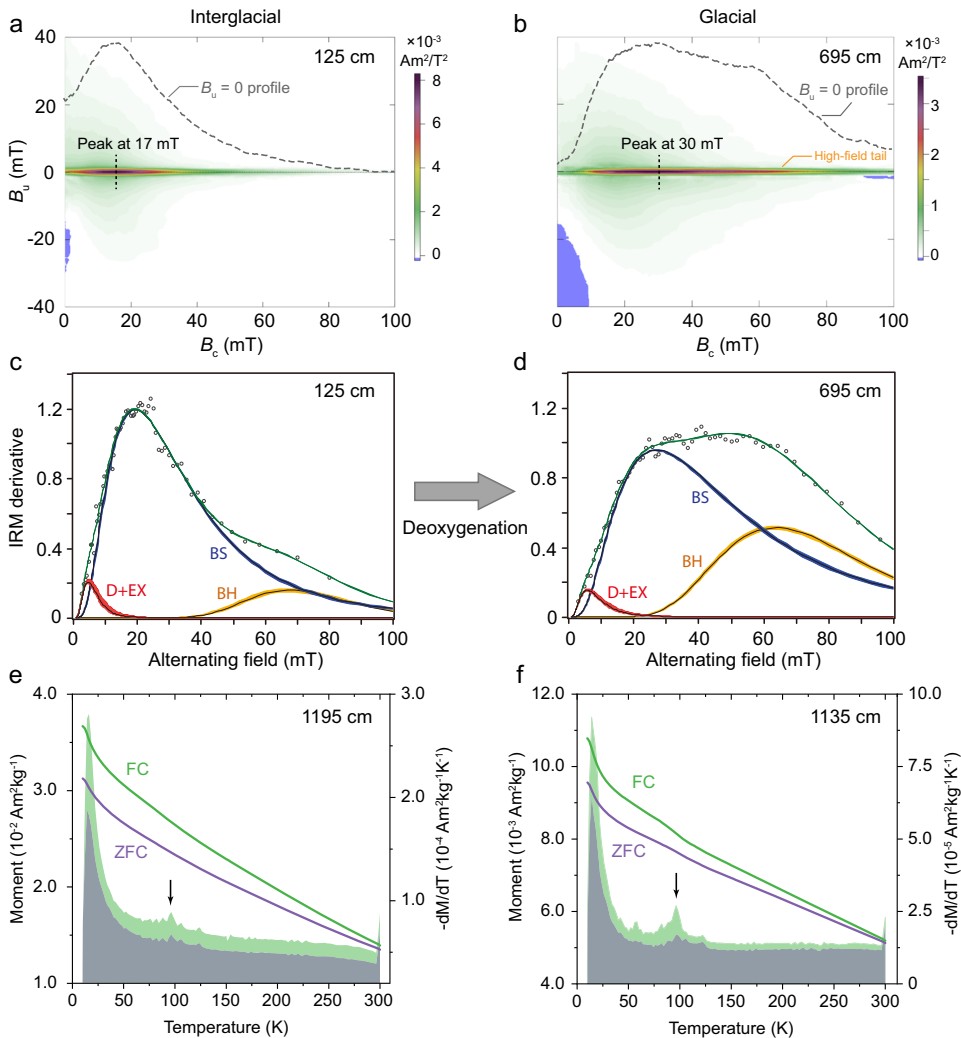

**Fig. 2 | Rock magnetic signatures of typical interglacial and glacial magnetofossil-bearing samples. a**, **b** First-order reversal curve (FORC) diagrams, **c**, **d** fitting of alternating field (AF) demagnetization data for saturation isothermal remanent magnetization (SIRM), and **e**, **f** low-temperature SIRM warming curves and associated derivatives after zero-field-cooled (ZFC) and field-cooled (FC) treatments. VARIFORC[64] smoothing parameters {$s_{c0}$, $s_{c1}$, $s_{b0}$, $s_{b1}$, $\lambda_c$, $\lambda_b$} are {7, 7, 3, 7, 0.1, 0.1} and {9, 9, 3, 9, 0.1, 0.1} in **a** and **b**, respectively. Horizontal profiles along $B_u = 0$ are shown in the FORC diagrams (dashed lines). In the IRM fitting curves, open circles are experimental data, the green line is the total fit, and red, blue, and orange lines are the three fitted log-Gaussian components. Shaded color areas represent error envelopes of 95% confidence intervals calculated using a resampling routine. D + EX (detrital + extracellular magnetite), BS (biogenic soft), and BH (biogenic hard) components are indicated in the coercivity spectra. Black arrows in **e**, **f** indicate the 100 K Verwey transition temperature associated with magnetofossils, which is less pronounced in the interglacial samples due to magnetite oxidation.

indicate that the glacial sample contains more magnetofossils with larger size and elongation (Fig. 3c, d, g, h). A Kolmogorov–Smirnov test based on empirical probability distributions confirms that the two magnetofossil populations are statistically distinct ($p < 0.001$, Fig. 3i). Violin plots of the length (Fig. 3j) and axial ratio (Fig. 3k) of magnetofossil crystals excluding bullets indicate clear trends from the interglacial to glacial stage: magnetofossil length increases, and axial ratio decreases. This trend is also clearly seen in the probability difference of the size distributions (length and width) between glacial and interglacial samples (Fig. 3l).

**Down-core profiles**
We use the BH magnetofossil fraction ($\delta_{BH}$; the ratio of the BH magnetofossil component to the sum of BS magnetofossil and BH magnetofossil components[22]) to reflect the 'magnetic hardness' of magnetofossils. Planktonic $\delta^{18}O$[34–37] (Fig. 4a), magnetic mineral concentration (saturation IRM (SIRM)) (Fig. 4b), bulk X-ray fluorescence (XRF) Ti/Ca ratio (Fig. 4c), $\delta_{BH}$ record (Fig. 4e), and additional magnetic and bulk geochemical records (Supplementary Fig. S4) are plotted

together for the upper ~20 m (~900 ka) of core MD00-2361, which indicate clear glacial-interglacial cycles (Fig. 4). Changes in bulk chemistry (Fig. 4c; Supplementary Fig. S4) reflect pronounced glacial-interglacial cyclicity of lithological and environmental changes linked to eolian activity and monsoonal precipitation in northwestern Australia[36,37]. Clear magnetic hardening of magnetofossils (i.e., an increased $\delta_{BH}$) is observed during glacials compared to interglacials (Fig. 4e). In addition, we find an overall large decrease in the total abundance of BS and BH magnetofossils during glacials compared to interglacials, with a large decrease in the absolute BS abundance and a slight increase in the BH abundance (Supplementary Fig. S5a, b).

Magnetofossils are preserved only above 17.55 m in the studied core (Supplementary Fig. S4). Bulk SIRM and ARM intensities drop by approximately one order of magnitude below ~17.55 m (Supplementary Fig. S4), which coincides with apparent magnetic property changes caused by reductive diagenesis, with a general XRF S content increase and the presence of pyrite nodules below this depth. FORC diagrams and IRM decomposition analyses for samples below 17.55 m indicate an apparent disappearance of the magnetofossil signature

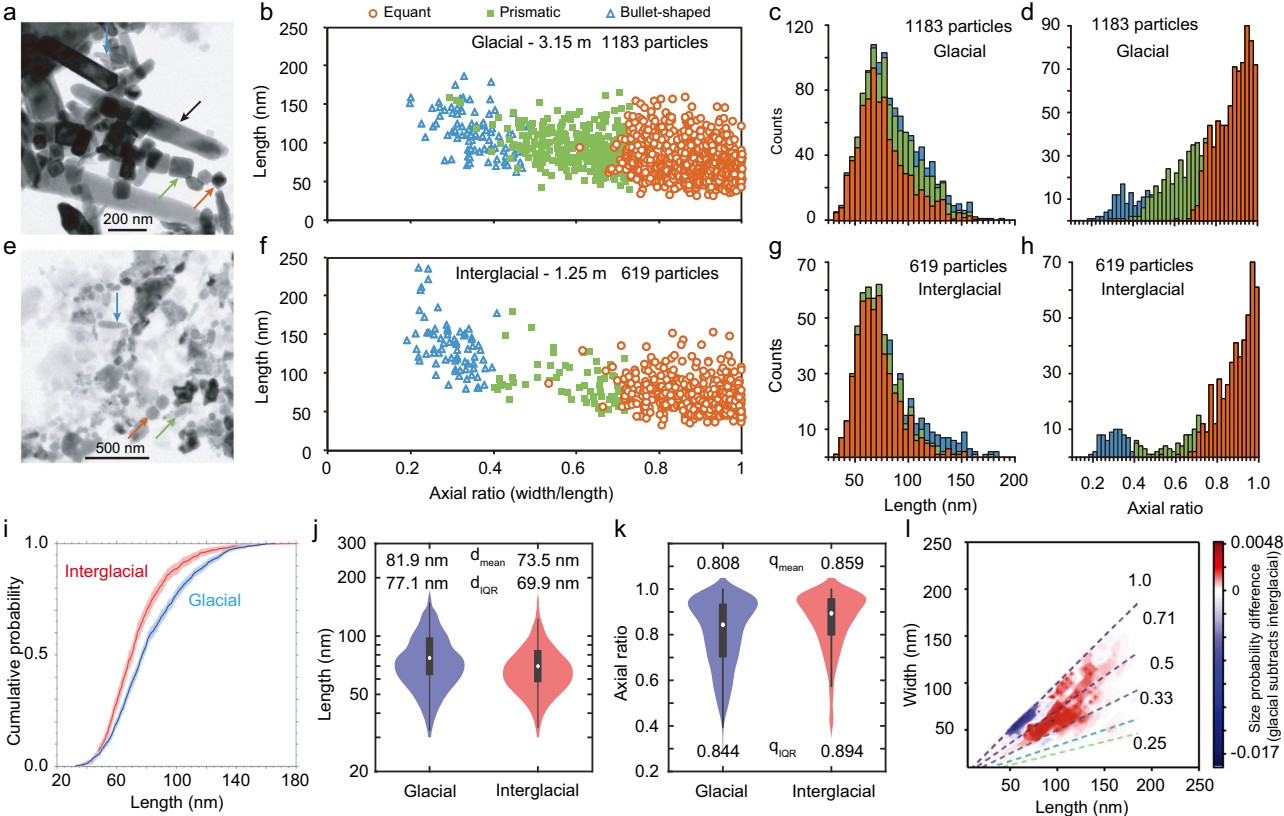

**Fig. 3 | Transmission electron microscope (TEM) observations and statistical analyses of magnetofossil morphology distributions. a, e** Bright-field TEM images. Typical magnetofossil morphologies, including more equant octahedral and cubo-octahedral crystals, elongated prismatic crystals, and bullet-shaped crystals are indicated by orange, green, and blue arrows, respectively. The black arrow indicates a larger elongated carbonate. **b, f** Length–axial ratio (width/length) plots for magnetofossil crystals measured from TEM images. Magnetofossil morphologies are categorized into three main groups: isotropic or nearly isotropic octahedral and cubo-octahedral crystals (orange open circles), elongated prismatic crystals (green solid squares), and bullet-shaped crystals (blue open triangles). The number of counted magnetofossil crystals is indicated. **c, d, g, h** Histograms of length and axial ratio for all counted magnetofossils. Orange, green and blue corresponds to nearly equant crystals, elongated prismatic crystals, and bullet-shaped crystals, respectively. **i** Empirical cumulative probability distribution of magnetofossil lengths, excluding bullet-shaped crystals for the two samples with Kolmogorov–Smirnov test results. **j, k** Violin plots of length and axial ratio of magnetofossils excluding bullet-shaped crystals. **l** Probability difference in size distributions (length and width) between the two glacial and interglacial samples. The dashed lines in **l** represent different aspect ratios. The two typical sediment samples are from **a–d** glacial (3.15 m) and **e–h** interglacial (1.25 m) intervals in core MD00-2361. The stratigraphic position for the two studied TEM samples is indicated in Supplementary Figure S4.

(Supplementary Fig. S1j) compared to those above (Supplementary Fig. S1a–i), due to sulfate-reducing diagenetic magnetofossil dissolution[24].

## Stable isotopes of benthic foraminifera

Twenty-five samples from several glacial and interglacial stages in core MD00-2361 were sieved to search for relevant benthic foraminiferal species (epifaunal *Cibicidoides wuellerstorfi* and deep infaunal *Globobulimina* spp.) required for the $\Delta\delta^{13}C$ BWO reconstruction. While *C. wuellerstorfi* was abundant, *Globobulimina* spp. was only found in four samples. Stable isotope and $\Delta\delta^{13}C$ proxy data (details can be found in Supplementary Table S2) are plotted along with the magnetofossil $\delta_{BH}$ record (Fig. 4e). For the two interglacial samples (at 1232.5 cm, MIS 11; 1630 cm, MIS 19), the $\Delta\delta^{13}C$ calibration function[11] gives oxygen concentration $[O_2]$ of ~150 and ~123 μmol/kg, respectively (Fig. 4e). These reconstructed interglacial $[O_2]$ values are in a similar range as modern $[O_2]$ near the studied core site (Fig. 1b). For the two glacial samples (at 280 cm, MIS 3; 1100 cm, MIS 10), the $\Delta\delta^{13}C$ calibration[11] gives a bottom-water $[O_2]$ of ~58 and ~115 μmol/kg, respectively (Fig. 4e). The reconstructed glacial low $[O_2]$ value at MIS 3 is consistent with the extremely high $\delta_{BH}$ during this glacial interval (Fig. 4e). The difference in the reconstructed $[O_2]$ value for the paired interglacial and glacial samples is ~50 μmol/kg, which is larger than the typical uncertainty of 17 μmol/

kg associated with the $\Delta\delta^{13}C$ calibration[11]. This indicates that the reconstructed $[O_2]$ values for the glacial and interglacial stage are distinct (Fig. 4e).

## Redox-sensitive trace-metal concentrations

Twenty samples were selected from the upper, middle, and lower parts of core MD00-2361 for redox-sensitive trace-metal concentration measurements as bottom-water oxygenation proxies. The samples cover several glacial-interglacial cycles: MIS 3–5, MIS 9–12, and MIS 19–21. Redox-sensitive authigenic uranium (aU)[8,15] and normalized metal data, such as Cd/Al, Mo/Al, and U/Al (ref. 8), are presented in Supplementary Table S3 and are plotted in Fig. 4e, f. Results indicate a pronounced glacial-interglacial contrast in redox-sensitive metal concentrations, i.e., higher aU, Cd/Al, Mo/Al, and U/Al values for glacial samples across all the studied glacial-interglacial cycles. These geochemical oxygenation proxy data have a consistent glacial-interglacial pattern with the magnetofossil $\delta_{BH}$ and benthic foraminiferal $\Delta\delta^{13}C$ records (Fig. 4e, f).

## Discussion

Previous magnetofossil-based studies have focused mostly on either bulk magnetic properties, or TEM observations of magnetofossil morphologies[20–23]. Here we combine magnetic analyses (Fig. 2),

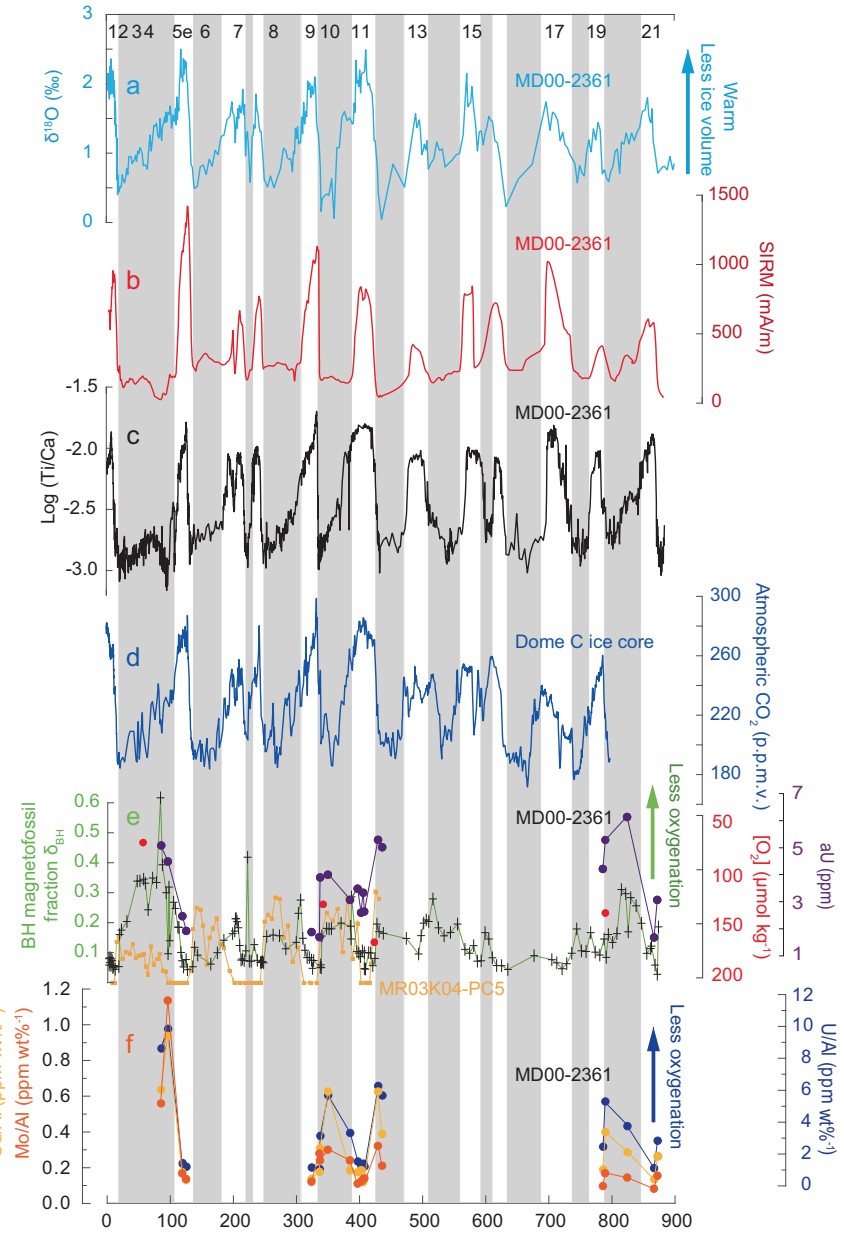

**Fig. 4 | Down-core profiles over the last 900 ka for core MD00-2361.**
**a** Planktonic $\delta^{18}O$ (refs. [34–36]). Odd numbers in **a** indicate interglacial periods.
**b** Bulk saturation isothermal remanent magnetization (SIRM) intensity from
u-channel data. **c** Bulk X-ray fluorescence (XRF) scanning records of log (Ti/Ca)
(ref. [37]). **d** Atmospheric $CO_2$ concentration reconstructed from Antarctic ice
cores[49]. **e** Relative BH (biogenic hard) magnetofossil fraction $\delta_{BH}$ (ratio of BH
content and the sum of BS (biogenic soft) and BH contents), benthic foraminiferal
$\Delta\delta^{13}C$ (red solid circles), and redox-sensitive authigenic uranium (aU; purple solid

circles) bottom-water oxygenation (BWO) proxy data for selected samples. Small
orange solid circles are the $\delta_{BH}$ record from Southern Ocean sediment core
MR03K04-PC5 (ref. [22]). **f** Normalized redox-sensitive trace-metal concentration
BWO proxy data of selected sediment samples: Cd/Al (orange solid circles), Mo/Al
(red solid circles), and U/Al (blue solid circles). Gray bars mark intervals with lower
planktonic $\delta^{18}O$ and XRF log (Ti/Ca) values for core MD00-2361, which broadly
correspond to glacial periods.

microscopic observations (Fig. 3), and micromagnetic simulations
(Supplementary Text S3, Figure S6)[25,38], which provide consistent
results that enable robust quantification of magnetofossil assemblage
changes through interglacial-glacial intervals in core MD00-2361. The
BH magnetofossil fraction $\delta_{BH}$ was used to trace redox changes across
Pleistocene glacial-interglacial cycles[22,23] and other paleoclimatic
intervals[25,39]. High $\delta_{BH}$ values correspond to a greater proportion of BH
magnetofossils and can reflect lower oxygenation because the BH
component (often associated with more elongated magnetite crystals)
reflects less-oxygenated environments compared to lower coercivity
more equant magnetite grains. Low-temperature magnetic data also

indicate that biogenic magnetite crystals in the glacial samples are
overall less oxidized than interglacial samples. These relationships
between magnetofossil morphology and oxygenation have been
documented commonly in both geological records[20–23,39–41] and
laboratory cultures[42–45], although more studies are needed to establish
links between different types of magnetofossil morphologies and
environmental conditions. For example, MTB strain MV-1 contains
elongated prismatic magnetic nanoparticles that grow strictly in
microaerobic and anaerobic environments[42,43,46]. For MTB model
strains AMB-1 and MSR-1 that biomineralize octahedral and cubo-
octahedral particles, pronounced magnetite particle size and

elongation increases occur when grown in environments with decreasing $O_2$ due to suppression of biogenic magnetite growth under more oxic environments[44,45]. Larger and more elongated magnetite crystals (i.e. larger $\delta_{BH}$) with less oxidation have also been reported in Eocene pelagic sediments, which were interpreted to reflect less-oxic environments[39]. Yamazaki and Kawahata[47] demonstrated a link between magnetofossil morphology and organic carbon flux in marine sediments, where more anisotropic crystals form preferentially in less-oxic and more organic-rich environments, possibly mediated by oxygenation and nutrient conditions.

Our data reveal a clear BH enrichment (i.e., elevated $\delta_{BH}$ values) from interglacial to glacial intervals (Figs. 2–4). Statistical analyses were made to quantitatively evaluate correlations between $\delta_{BH}$ and other proxy data (Supplementary Figure S7). For our data from the last 21 marine isotope stages, increased $\delta_{BH}$ is observed for most glacial periods, except for a few intervals with unclear patterns, e.g., MIS 6 (Supplementary Figure S7c, d). We also note that not all glacial intervals have large BH magnetofossil increases. For example, MIS 6 and 16 have low $\delta_{BH}$ values of ~0.1 (Fig. 4e). We suspect that variable oxygenation and organic carbon supply, as well as other complex paleoceanographic conditions, may have produced variable $\delta_{BH}$ increases over different glacial events.

Decreased pore water oxygenation close to the water/sediment interface during glacial stages, through diffusion of decreased BWO, could have stimulated biomineralization of larger and more elongated magnetite crystals, although there may be a minor contribution from organic matter decomposition. This glacial BWO concentration decrease for the studied core is supported by several lines of evidence. (1) Our long magnetofossil BWO reconstruction is validated directly by the quantitative $\Delta\delta^{13}C$ (Fig. 4e), aU (Fig. 4e), and redox-sensitive trace-element concentration BWO proxies (Fig. 4e, f) for typical glacial/interglacial samples across MIS 3–5, MIS 9–12, and MIS 19–21, where the reconstructed glacial BWO concentration during glacial periods (MIS 3, 4, 10, 12, 20) is consistently lower than interglacial periods (MIS 5e, 9, 11, 19, 21). (2) Biogenic magnetite crystals in glacial samples are less oxidized, which reflects less-oxygenated environments for magnetite biogenesis. (3) Sediment color for the glacial intervals is beige/greenish compared to brown/reddish interglacial intervals, which likely reflects lower and higher BWO conditions, respectively (Supplementary Fig. S4). (4) A magnetofossil record for a Southern Ocean core MR03K04-PC5 (41°33.07′S, 90°24.39′E; 2913 m water depth; Fig. 1) has a similar glacial enrichment in the BH magnetofossil fraction and more anisotropic magnetofossil crystals[22] as observed in our record (Fig. 4e). (5) Down-core magnetofossil and geochemical variations are synchronous with glacial-interglacial cyclicity (Fig. 4; Supplementary Fig. S4), with any offsets no larger than the sampling resolution (10-cm). This indicates that biogenic magnetite biomineralization and surface ocean variations are likely associated with climatic conditions rather than resulting from redox reactions much deeper within the sediment, which would create an apparent signal offset (e.g., with respect to planktonic $\delta^{18}O$). The evidence consistently indicates a large-scale deep-sea deoxygenation pattern in the glacial Indian Ocean. Bottom ocean currents are suggested to have decreased during glacials near the studied area; for example, Murgese and De Deckker[48] interpreted low glacial carbon isotope values of the benthic foraminifera species *Cibicidoides wuellerstorfi* in the eastern Indian Ocean to reflect reduced glacial deep-water ventilation. Reduced circulation would bring less nutrients for magnetite biomineralization. Geochemical records for the studied core also indicate decreased nutrient supply associated with decreased glacial monsoonal precipitation[36,37]. This is consistent with the observation that the total magnetofossil content (BS and BH) decreases during glacials compared to interglacials (Supplementary Figure S5). Thus, we propose that, the combination of proxy results of increased magnetofossil $\delta_{BH}$, decreased benthic foraminiferal $\Delta\delta^{13}C$, and increased redox-sensitive

metal concentrations (Fig. 4e, f) during glacials compared to inter-glacials, are mainly controlled by decreased seawater oxygenation, while increased organic carbon and nutrient supply may only make a minor contribution at the studied core site.

Statistical analyses of the correlations between proxy data sets (Supplementary Figure S7a, c, e) indicate that the magnetofossil $\delta_{BH}$ record of BWO in the eastern Indian Ocean over the last 850 ka correlates with atmospheric $CO_2$ variations recorded in Antarctic ice cores[49], where glacial low-$CO_2$ intervals correspond to deoxygenation periods (Fig. 4d, e; Supplementary Figure S7c). Oxygen concentrations in seawater are influenced by various factors that control its supply and consumption. At the ocean surface, oxygen concentrations are close to saturation values, through wind-driven air-sea oxygen exchange. Such exchange is modulated by temperature and salinity, which affect oxygen solubility[50]. $O_2$ is also produced by photosynthesis (whereas $CO_2$ is consumed) in the ocean surface layer. Oxygen utilization is controlled mainly by organic matter respiration throughout the water column. Climate models predict ocean deoxygenation with ongoing global warming through a combination of lower oxygen solubility, increased oxygen consumption due to decreased ventilation, and oceanic circulation changes[1,50].

In contrast, due to lower seawater temperatures, oxygen solubility (oxygen saturation concentration) during glacial periods would have been higher, while it has been postulated that organic matter remineralization rates were reduced[51], potentially leading to increased upper ocean (e.g., > 1.5 km) oxygen contents, for example, during the Last Glacial Maximum in the Indian Ocean[50]. For the glacial deep oceans, the amount of cooling-driven $O_2$ increase (through increased oxygen solubility) did not keep up with the amount of oxygen lost through oxygen utilization/ventilation, which led to a deep-water oxygen concentration decrease. It is thought that the decreased deep-sea oxygen exposure may have led to increased organic carbon burial during glacial maxima[52]. Consistent documentation of lower deep-sea oxygen contents from carbon isotope and geochemical records[7–9,11–15] in the Atlantic and Pacific Oceans, and a late Pleistocene magnetic mineral dissolution-based redox proxy record from the northwest Pacific Ocean[10], are regarded as evidence of increased respired carbon storage that could have contributed to lower atmospheric $CO_2$ concentrations recorded in ice cores[49] during glacial periods. Furthermore, modeling studies suggest that increased glacial air-sea disequilibrium may have also driven glacial $CO_2$ drawdown[53].

Our magnetofossil record indicates significant glacial Indian Ocean deep-water deoxygenation (Fig. 4) that persisted during glacial periods over the last 850 ka. The same pattern is obtained through quantitative BWO reconstruction using the benthic foraminiferal $\Delta\delta^{13}C$ proxy and redox-sensitive trace-metal reconstruction on samples from the same core through several glacial and interglacial intervals. Consistent glacial and interglacial BWO cycles for a broader geographic area in the Indian Ocean can also be inferred from fragmented published records. A similar glacial $\delta_{BH}$ enrichment in Southern Indian Ocean core MR03K04-PC5[22] suggests decreased deep-sea oxygenation during glacials (Fig. 4e) over a geographically large Indian Ocean area (Fig. 1). In eastern Indian Ocean core BMR 96GC21 (23°46.33′S, 108°30.04′E; 2100 m water depth) near our studied site (Fig. 1), $\delta^{13}C$ and benthic foraminifera assemblage data indicate more negative $\delta^{13}C$ values and higher *U. peregrina* and *U. proboscidea* species counts during MIS 2 and 6[54]. In eastern Indian Ocean core Fr10/95 GC5 (14°00.55′S, 121°01.58′E; 2400 m water depth; Fig. 1) and Ocean Drilling Program (ODP) Hole 760 A (16°55.32′S, 115°32.48′E; 1970 m water depth; Fig. 1), generally low glacial carbon isotope values of the benthic foraminifera species *Cibicidoides wuellerstorfi* reflect reduced glacial deep-water ventilation[48,54–56]. These magnetofossil records and benthic foraminifera records indicate consistently less-oxic glacial BWO conditions over a large area in the Indian Ocean (Fig. 5). The lower glacial BWO concentrations in the

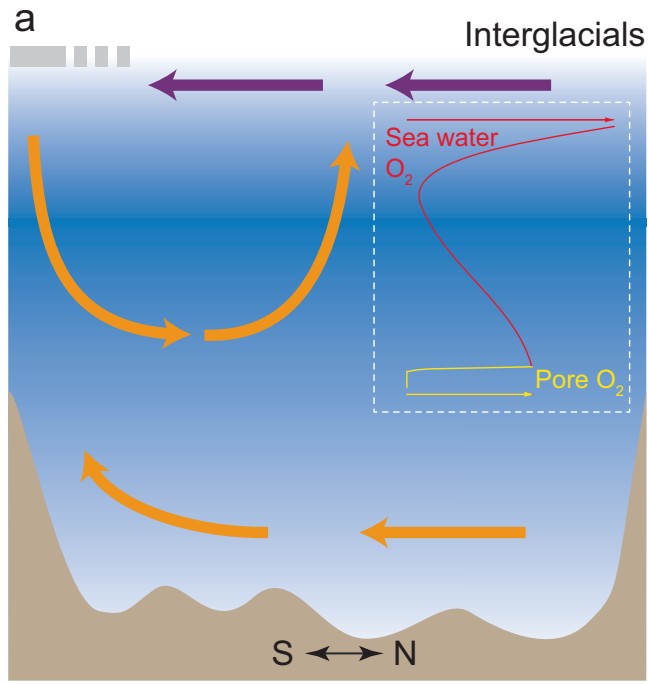

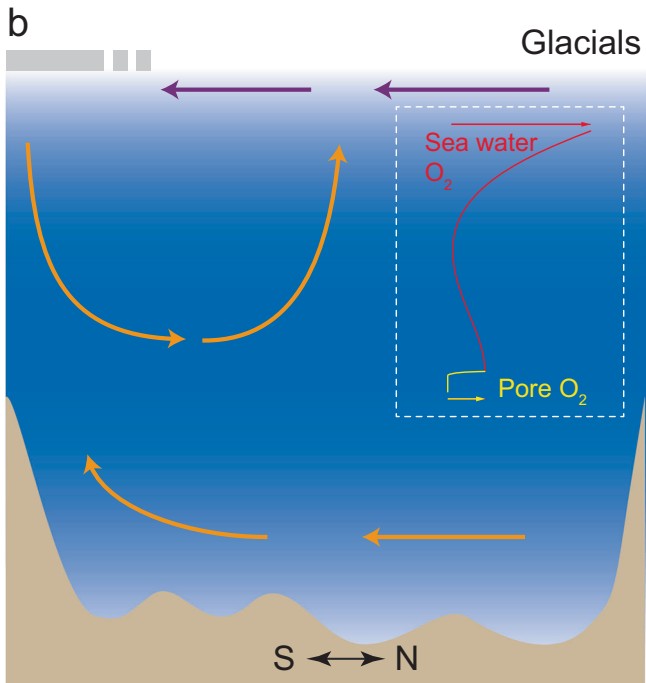

**Fig. 5 | Conceptual models of eastern Indian Ocean oxygenation evolution during recent glacial/interglacial cycles. a** Interglacial stages. Stronger over-turning circulation, enhanced deep-water ventilation, better-oxygenated conditions, and lower respired carbon storage. **b** Glacial stages. Weaker overturning circulation, reduced deep-water ventilation, less-oxygenated conditions, and expanded respired carbon reservoir. Inset (dashed white boxes): schematic illustration of seawater $O_2$ profiles for the entire water column (red) and uppermost sediment pore water (yellow). The depth scales are exaggerated. Purple and orange arrows indicate surface-water and deep-water flows, respectively. Gray bars on the top left-hand side indicate the extent of sea-ice cover and polynyas. Ocean processes in this figure are described according to the "nutrient deepening" hypothesis of Boyle[5,6], and follow previous illustrations[2,9].

eastern Indian Ocean are unlikely related to expansion of low-$O_2$ water from the Bay of Bengal oxygen minimum zone (OMZ) because of their large geographic separation (Fig. 1a, b).

Moreover, compilation of sediment oxygenation proxy data[4,57] and ocean modeling results[50] consistently reveal a clear contrast between the upper and deep Indian Ocean from the last glacial maximum (LGM) to the mid-Holocene (MH), with a general deoxygenation trend in the upper ocean and oxygenation of the deeper ocean. The simulated deep ocean gains oxygen from the LGM to the MH, with a mean 15.1 µmol m$^{-3}$ increase when averaged over 2000–5000 m (ref. 50). Our long magnetofossil record, benthic foraminiferal $\Delta\delta^{13}C$, and redox-sensitive trace-metal concentration reconstructions of deglaciation oxygenation changes are broadly consistent with the sediment proxy compilation and ocean modeling results from the LGM to MH.

Our high-resolution magnetofossil record over long glacial-interglacial cycles provides evidence of a significant glacial deep-water oxygenation decline in the eastern Indian Ocean through at least the last 850 ka. Decreased glacial South Indian Ocean deep-water oxygenation more likely reflects higher respiration rates due to enhanced organic carbon storage in the ocean water column combined with reduced ventilation (possibly due to sea-ice expansion and reduced episodic opening of polynyas, and stronger haline stratification). All of these conditions would have contributed to an accumulated ocean carbon reservoir (Fig. 5). Similar scenarios have been demonstrated across the last deglaciation in the Pacific and Atlantic Oceans[11–16,57–59]. These data strongly support the hypothesis of ocean carbon storage as a major mechanism for regulating glacial-interglacial $CO_2$ variations. We show that the Indian Ocean likely contributed significantly to lower glacial atmospheric $CO_2$ concentrations over recent glacial/interglacial cycles.

In summary, magnetic measurements from a marine sediment core from offshore of Western Australia reveal a magnetofossil record that spans the last 21 MIS, where magnetofossil coercivity is enhanced during glacial periods. Such magnetic property variations are explained by an enriched elongated and also less oxidized magneto-fossil component, which consistently suggests lower glacial bottom-water oxygenation during glacial periods that enabled biogenesis of larger and more elongated biogenic magnetite crystals, although a minor contribution from changes in organic matter and nutrient supply cannot be ruled out. Our magnetofossil record of lowered deep-sea oxygenation is consistently supported by redox-sensitive trace-element concentration and quantitative benthic foraminiferal $\Delta\delta^{13}C$ proxy data from the same core, and is also consistent with magnetofossil proxy data from a core in the southern Indian Ocean. Our multi-proxy indication of lowered deep-sea oxygenation linked to carbon reservoir build-up during glacial periods in the eastern Indian Ocean, which occurred persistently at least for the last 850 ka, supports the hypothesis that increased respired carbon storage drove lower atmospheric $CO_2$ concentrations. Our results also demonstrate the potential of magnetofossils as a sensitive proxy for tracing redox conditions to understand carbon cycle dynamics over climate transitions.

## Methods

### Marine sediment materials and age model
Marine sediment core MD00-2361 was recovered from offshore North West Cape, Western Australia (113°28.63′E, 22°04.92′S; 42 m length; 1805 m water depth; Fig. 1). The core site lies below the present-day shallow water Leeuwin Current (LC), which is initiated from two sources: the Indonesian Throughflow and the Indian Ocean East Gyre Current (EGC). The LC overrides the equator-moving West Australian

Current[56] (WAC; Fig. 1). The South Indian Ocean deep circulation is characterized by the eastward-flowing Antarctic Circumpolar Current (ACC) and the gyre in the central Indian Ocean with offshoots emanating from the ACC[60,61] (Fig. 1). Interglacial and glacial intervals alternate between sediments that are brown and rich in fluvially-supplied detritus and beige marine pelagic carbonates with enriched eolian detritus, respectively[34,36,37,62]. The Matuyama-Brunhes boundary (~780 ka) is identified at ~16.2 m in core MD00-2361[35]. Planktonic $\delta^{18}O$ data for the upper 13.6 m of this core supplemented with additional analyses back to MIS 21[37] indicates that the core records continuous sedimentation. There is a small disagreement within MIS 5–4 between the $\delta^{18}O$ and XRF records, where $\delta^{18}O$ has a typical MIS 5 pattern, but terrigenous elements in the XRF data reach a minimum after MIS 5e. The age model is defined based on the planktonic $\delta^{18}O$ record. Paleomagnetic data and preliminary magnetic results from this core were reported by Heslop et al.[30,35], who showed that magnetofossils are abundant.

## Rock magnetism

Subsamples were taken from u-channels at 10 cm stratigraphic intervals from the upper 20 m of core MD00-2361. Hysteresis, IRM acquisition curves, and FORC measurements were made with a Princeton Measurements Corporation vibrating sample magnetometer (VSM; model 3900) at the Research School of Earth Sciences, Australian National University (ANU). Hysteresis loops were measured between −500 and +500 mT with a field step of 5 mT and 350 ms averaging time. FORC diagrams[27] were measured for ~20 samples, which were obtained with a 1 T maximum applied field, $B_u = [−50\,mT, 50\,mT]$, $B_c = [0, 100\,mT]$, with 200 FORCs measured (corresponding to a field spacing of ~1 mT), and 350 ms averaging time. FORC diagrams were processed using the FORCinel software version 3.06[63] with the VARIFORC protocol[64].

Most glacial carbonate-rich samples are magnetically weak and have noisy IRM derivative curves when estimated from VSM measurements. Therefore, we performed coercivity spectrum measurements using a cryogenic magnetometer (2-G Enterprises 755 R) at ANU. All samples ($n = 152$) were subjected to AF demagnetization of an ARM and IRM to determine coercivity distributions. Samples were first subjected to three-axis demagnetization with a 140 mT AF. An ARM was imparted to samples using a 0.05 mT direct current (DC) bias field superimposed on a decaying 140 mT peak AF. Samples were then subjected to stepwise AF demagnetization with 55 logarithmically spaced field steps to a maximum field of 140 mT; ARM intensities were measured after each AF step. Then, an IRM was imparted to samples at 1 T using a 2-G Enterprises pulse magnetizer. The IRM was subjected to stepwise AF demagnetization using the same steps and procedure as ARM measurements. Coercivity distributions were fitted with three logarithmic Gaussian distributions: two around the clear peaks and one around the low field region. Each identified component is defined by three parameters: a component percentage, a peak $B_c$ value, and a dispersion parameter[65].

For low-temperature magnetic measurements, samples were cooled from 300 K to 10 K at 12 K/min with both an absence (zero-field cooled, ZFC) and presence (field cooled, FC) of a 2.5 T field. A saturation remanent magnetization was imposed using a 2.5 T field at 10 K, and then the magnetization was measured in 2 K steps when heating from 10 K to 300 K at 2 K/min. Low-temperature measurements were performed with a Quantum Design Magnetic Property Measurement System (MPMS3) at the School of Physics, Peking University, Beijing.

## Magnetic extraction and transmission electron microscope observations

Two characteristic interglacial and glacial samples (taken from 1.25 and 3.15 m, respectively) were subjected to TEM analysis to obtain information about magnetofossil size and shape distributions. Magnetic extraction was performed with a Frantz isodynamic magnetic separator and a long glass tube with a stopcock at the base[66]. Magnetic extracts were viewed and analyzed using a Philips CM300 TEM operated at 300 kV at ANU. TEM images containing magnetofossils were acquired randomly. All magnetofossil crystals in the TEM images were counted to prevent biased BS and BH estimates.

## Micromagnetic simulations

The link between magnetic properties and magnetofossil morphology was investigated using a micromagnetic model of magnetofossil ensembles of stable SD particles. Simulations were performed with a modified version of the FORCulator software[67], which enables magnetic property simulations for magnetofossil ensembles with realistic size and shape distributions[38].

## Benthic foraminiferal $\Delta\delta^{13}C$ and $\delta^{18}O$

Benthic foraminifera $\delta^{18}O$ and $\delta^{13}C$ were analyzed at the British Geological Survey (BGS) National Environmental Stable Isotope Facility using an IsoPrime dual inlet mass spectrometer plus Multiprep device. Isotope values ($^{13}C$, $^{18}O$) are reported as per mille (‰) deviations of the isotopic ratios ($^{13}C/^{12}C$, $^{18}O/^{16}O$) calculated to the VPDB scale using a within-run laboratory standard (KCM long-term average $+2.00 \pm 0.05\%$ $\delta^{13}C$, and $−1.73 \pm 0.05\%$ $\delta^{18}O$) calibrated against international standard NBS-19. The $\Delta\delta^{13}C$ gradient between epifaunal *C.wuellerstorfi* and infaunal species *Globobulimina* spp. quantitatively reflects BWO concentrations, and follows a linear relationship where $\Delta\delta^{13}C = 0.0064 \times [O_2] + 0.555$ ($55 < [O_2] < 235\,\mu mol/kg$)[11].

## Geochemical measurements

Bulk chemical compositions were analyzed on archive cores at 1-cm resolution using an Avaatech XRF core scanner at the Royal Netherlands Institute for Sea Research. Some of the XRF data for core MD00-2361 were presented by Stuut et al.[36,37], who described detailed measurement procedures.

Quantitative geochemical compositions were determined via inductively coupled plasma-optical emission spectroscopy (ICP-OES) and mass spectrometry (ICP-MS) at the State Key Laboratory of Marine Geology, Tongji University (China). Approximately 50-mg of powdered sample was digested with a mixture of $HNO_3 + HF$ on a hot plate. Subsequently, the eluted samples were diluted by 2% $HNO_3$ for major- and trace-element measurements by ICP-OES (IRIS Advantage) and ICP-MS (Thermo VG-X7), respectively. Uncertainties were monitored by replicate analyses of BHVO-2, W-2a, GSP-2, and GSD-9 standards, with relative deviations < 5% for the reported data.

Authigenic uranium (aU) is the insoluble U (IV) form precipitated in the sediment, reducing from soluble U (VI) from the pore water when oxygen concentrations decrease or organic flux increases[15]. The aU concentration can be calculated as: $^{238}U_{authigenic} = {}^{238}U_{total} − 0.4 \times {}^{232}Th$, where $^{232}Th$ is completely delivered from detrital $^{238}U$ and follows the activity ratio ($0.4 \pm 0.1$) in the Indian Ocean[68]. Though influenced by two factors, aU reflects qualitative bottom-water oxygenation changes over glacial-interglacial cycles[15]. Other redox-sensitive metals (e.g., Cd, Mo) have a similar reaction process with U, despite differing in reaction details[69], and reflect the degree of deoxygenation. Normalized metals (e.g., metals/Al) are used to eliminate the effect of sedimentation rate variations, which influence both metals and lithogenic elements such as Al. Therefore, the effect is counteracted.

## Data availability

The rock magnetic experimental data, micromagnetic simulation results, and geochemical data that support the findings of this study are available in Zenodo and can be accessed at https://doi.org/10.5281/zenodo.8002889, as well as in Supplementary Tables 1–3.

## Code availability

The micromagnetic simulation code related to this study is available in Zenodo with the accessible link https://doi.org/10.5281/zenodo.8002869.

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

## Acknowledgements

This study was supported by the National Natural Science Foundation of China (NSFC grant 41974074 to L.C.), a Royal Society-Newton Advanced Fellowship (NAF\R1\201096), and a joint NSFC grant (42061130214) to L.C. and R.J.H., the UK Research and Innovation (UKRI Future Leaders Fellowships grant MR/S034293/1 to B.A.A.H.), and the Australian Research Council (ARC grants DP160100805 and DP200100765 to A.P.R. and D.H.).

## Author contributions

L.C. conceived and designed the research, performed the magnetic measurements and microscopic observations, and drafted the manuscript. B.A.A.H. and M.L. conducted benthic foraminifera stable isotope analysis. L.C., X.Z., P.X., and T.A.B. performed coercivity distribution analysis. L.C., R.H., and B.H. contributed to trace-metal concentration analysis. L.C. and R.J.H. performed micromagnetic simulations. Z.P., L.C., X.Z., and D.H. contributed to statistical analysis. F.Z. performed magnetofossil counting. S.W. conducted low magnetic measurements. L.C., B.A.A.H., D.H., A.P.R., and P.D.D. contributed to data interpretation. L.C., B.A.A.H., D.H., X.Z., A.P.R., P.D.D., P.X., Z.P., F.Z., R.H., B.H., S.W., T.A.B., M.L., J.B.W.S., and R.J.H. contributed to the review and edit of the manuscript.

## Competing interests

The authors declare no competing interests.
