## [Peer Review File · Nature Communications]

Indian Ocean glacial deoxygenation and respired carbon accumulation during mid-late Quaternary ice agesREVIEWER COMMENTS

Reviewer #1 (Remarks to the Author):

The findings of this paper are not new but, as the authors note, this is the first evidence from the Indian Ocean (as far as I know) for lower oxygen concentrations in deep waters during glacial intervals, so the paper is worthy of publication. However, the conclusions could be much stronger, and the paper could have a much greater impact, if the authors cited the wide variety of methods that all indicate lower oxygen in the deep ocean during late Pleistocene glacial intervals. As already described in this paper, the oxygen concentration of the deep sea is directly related to carbon storage by the biological pump. The second paragraph of the main text provides a limited summary of some of the approaches that have been used to constrain past oxygen concentrations, but a much more thorough review of the evidence could be made here. Importantly, even though each proxy involves substantial assumptions and uncertainties, the variety of proxy methods that give consistent results showing lower oxygen concentrations throughout the deep ocean during glacial intervals affords great confidence to the findings. The conclusions of the paper would be much stronger if the authors emphasized this point about the consistency of diverse methods, and it could be done very concisely, just by listing in the main text the methods that indicate lower deep sea oxygen concentrations during glacial intervals (some references are provided below). The consistency among all of these records gives confidence in the conclusion that the climate-related cycle of atmospheric CO₂ reconstructed from ice cores is directly related to the amount of carbon stored in the deep ocean by the biological pump.

The most recent approach to reach these conclusions (that I am aware of) is to reinterpret the sub-surface manganese peaks, which have puzzled marine geochemists since at least the 1970s, as a further indicator of low oxygen concentrations in the deep ocean during glacial periods. Several recent papers have made this point, which has been summarized together with new data in a synthesis by Pavia et al [2021]. The current paper by Chang et al. would be much stronger if its summary included this evidence.

Figure 5 of the Chang et al paper is but one of many illustrations of the “nutrient deepening” hypothesis first posited by Boyle [1988a; b]. For example, illustrations similar to Fig 5 appear in in Sigman and Boyle [2000] (their Fig 6) and in Bradtmiller et al [2010] (their Fig 6, which more than Sigman and Boyle 2000 is similar to Fig 5 in current paper). I recommend that these papers be acknowledged in the caption of Chang’s Figure 5.

Lastly, the results of this paper appear to someone who is not an expert in paleomagnetic methods to be very similar to the results of Korff et al [2016], both with respect to the conclusions about the impact of changes in bottom-water oxygen on the magnetic properties of sediments, and with respect to the time interval covered by the records. The Korff paper is cited in the SI (ref 16) but I recommend that it be worked into main text. Chang et al may wish to describe similarities or differences in methods, but most importantly I recommend that they describe the consistent conclusions reached by the two papers.

To reiterate, the paper is in fine shape and could be published in its present form, but the paper could have much greater impact if it provided a more complete review of relevant literature. Nearly 3 pages of text are currently devoted to magnetic results. If Chang et al are limited by the length of the paper, then perhaps some of the text concerning magnetic results could be moved to the SI to make space to accommodate the reference citations that I believe will make this a stronger paper.

Key References:

Nutrient deepening hypothesis proposed - [Boyle, 1988a; b]

First paper to combine evidence from organic carbon flux proxy and sediment redox conditions to conclude that BWO must have been lower during the last glacial period – [Francois et al., 1997]

Clearest (in my opinion) explanation of the processes by which the ocean affects atmospheric CO₂, and graphical illustration of nutrient deepening – [Sigman and Boyle, 2000]

After a hiatus of more than a decade, the beginning of many studies that combined evidence from organic carbon flux proxy indicators and sediment redox conditions to conclude that BWO must have been lower during the last glacial period - [Jaccard et al., 2009] and [Bradtmitter et al., 2010]

Paper using magnetic properties of sediments to conclude that BWO was lower during glacial periods – [Korff et al., 2016]

Calibration and application of Dd13C of paired benthic foraminifera to conclude that BWO was lower under glacial conditions - [Hoogakker et al., 2015; Hoogakker et al., 2018]

Calibration and application of preservation of biogenic organic compounds to conclude that BWO was lower under glacial conditions – [Anderson et al., 2019]

Subsurface Mn peaks linked to lower BWO under glacial conditions - [Pavia et al., 2021]

Publications:

Anderson, R. F., J. P. Sachs, M. Q. Fleisher, K. A. Allen, J. Yu, A. Koutavas, and S. L. Jaccard (2019), Deep-Sea Oxygen Depletion and Ocean Carbon Sequestration During the Last Ice Age, *Global Biogeochemical Cycles*, 33(3), 301-317, doi:10.1029/2018GB006049.

Boyle, E. A. (1988a), The role of vertical chemical fractionation in controlling late Quaternary atmospheric carbon-dioxide, *Journal of Geophysical Research-Oceans*, 93(C12), 15701-15714.

Boyle, E. A. (1988b), Vertical oceanic nutrient fractionation and glacial interglacial CO₂ cycles, *Nature*,

331(6151), 55-56.

Bradt Miller, L. I., R. F. Anderson, J. P. Sachs, and M. Q. Fleisher (2010), A deeper respired carbon pool in the glacial equatorial Pacific Ocean, *Earth and Planetary Science Letters*, 299(3-4), 417-425, doi:<https://doi.org/10.1016/j.epsl.2010.09.022>.

Francois, R., M. A. Altabet, E. F. Yu, D. M. Sigman, M. P. Bacon, M. Frank, G. Bohrmann, G. Bareille, and L. D. Labeyrie (1997), Contribution of Southern Ocean surface-water stratification to low atmospheric CO₂ concentrations during the last glacial period, *Nature*, 389(6654), 929-935.

Hoogakker, B. A. A., H. Elderfield, G. Schmiedl, I. N. McCave, and R. E. M. Rickaby (2015), Glacial-interglacial changes in bottom-water oxygen content on the Portuguese margin, *Nature Geoscience*, 8(1), 40-43, doi:<https://doi.org/10.1038/ngeo2317>.

Hoogakker, B. A. A., Z. Lu, N. Umling, L. Jones, X. Zhou, R. E. M. Rickaby, R. Thunell, O. Cartapanis, and E. Galbraith (2018), Glacial expansion of oxygen-depleted seawater in the eastern tropical Pacific, *Nature*, 562(7727), 410-413, doi:10.1038/s41586-018-0589-x.

Jaccard, S. L., E. D. Galbraith, D. M. Sigman, G. H. Haug, R. Francois, T. F. Pedersen, P. Dulski, and H. R. Thierstein (2009), Subarctic Pacific evidence for a glacial deepening of the oceanic respired carbon pool, *Earth and Planetary Science Letters*, 277(1-2), 156-165, doi:<https://doi.org/10.1016/j.epsl.2008.10.017>.

Korff, L., T. von Dobeneck, T. Frederichs, S. Kasten, G. Kuhn, R. Gersonde, and B. Diekmann (2016), Cyclic magnetite dissolution in Pleistocene sediments of the abyssal northwest Pacific Ocean: Evidence for glacial oxygen depletion and carbon trapping, *Paleoceanography*, 31(5), 600-624, doi:<https://doi.org/10.1002/2015PA002882>.

Pavia, F. J., S. Wang, J. Middleton, R. W. Murray, and R. F. Anderson (2021), Trace Metal Evidence for Deglacial Ventilation of the Abyssal Pacific and Southern Oceans, *Paleoceanography and Paleoclimatology*, 36(9), e2021PA004226, doi:<https://doi.org/10.1029/2021PA004226>.

Sigman, D. M., and E. A. Boyle (2000), Glacial/interglacial variations in atmospheric carbon dioxide, *Nature*, 407(6806), 859-869.

Reviewer #2 (Remarks to the Author):

Please see the attached PDF of my review.

Reviewer #3 (Remarks to the Author):

The authors present a long magnetofossil record of bottom water oxygenation (BWO) in one sediment core from the eastern Indian Ocean. They interpret their record to reflect low BWO during glacial periods, which they believe suggests widespread carbon sequestration in the deep Indian Ocean (similar to what is thought for both the Pacific and Atlantic Oceans).

I do not think that the current paper can be published in Nature Communications. First, I must point out that I am not an expert in the magnetofossil field. However, if I assume that the interpretation of their proxy is correct, I still have a problem with the interpretation of the data as presented for two reasons:

1) Only 2 samples were analyzed using the $\Delta\delta^{13}\text{C}$ proxy to corroborate their magnetic fossil interpretation. The two samples were analyzed at the Stage 10/11 boundary. The fact that their attempt to make this measurement at one other boundary (5/6 boundary) was unsuccessful is disconcerting. A glacial-interglacial BWO interpretation at one climate boundary using the $\Delta\delta^{13}\text{C}$ proxy is not a significant line of evidence. You would need more than a pair of data points to convince me, and not only from the same boundary (10/11) but from several boundaries. I believe their $\Delta\delta^{13}\text{C}$ data set needs to be expanded.

2) No statistical analysis is presented comparing the interpreted deoxygenation record in Figure 4e with the atmospheric CO_2 record in Figure 4d (this, arguably, is the crux of the manuscript), even though the authors say that the records are correlated. Perhaps there is a relationship between glacials and enhanced deoxygenation but it needs to be made explicit in a quantitative way. Also, if there is a relationship, it breaks down sometimes. For example, Stage 6 has a low biogenic BH fraction (should be high according to their interpretation) and Stage 7 has a high biogenic BH fraction (should be low according to their interpretation). Another example that goes against what the authors claim: in Stage 2 biogenic BH fractions decrease as atmospheric CO_2 decreases (the reverse should be the case according to the authors).

Some other comments:

What is the significance of the Ti/Ca record? I'm surprised that the strong correlation between the Ti/Ca and climate is not discussed. What is the cause of the relationship? I think the authors should do more with this data. Also, how do the geochemical relationships in the XRF data (Mn/Ti, Ti/Ca, S/Ti) relate to redox proxies in the sediment post-depositionally. If you can dissolve the biogenic magnetite at levels below 1770 m, why can't something similar be happening in the sediment intervals above 1770 m? Or, another way of putting this, why is diagenesis different below 1770 at this site?

With respect to the bigger picture, other cores are implicated for having the same BH glacial enrichment, but none of the data from these other cores are shown in the figures. Why not present the data from the other cores to make your case stronger? For example, on lines 262-264 it's stated that the authors' record is broadly consistent with the sediment proxy compilation and ocean modeling results. Why can't the compilation be made in a figure to show the broad consistency? Otherwise, the reader cannot assess the broad consistency claim that is made.

Response to Reviewer Comments

Reviewer #1 (anonymous)

The findings of this paper are not new but, as the authors note, this is the first evidence from the Indian Ocean (as far as I know) for lower oxygen concentrations in deep waters during glacial intervals, so the paper is worthy of publication. However, the conclusions could be much stronger, and the paper could have a much greater impact, if the authors cited the wide variety of methods that all indicate lower oxygen in the deep ocean during late Pleistocene glacial intervals. As already described in this paper, the oxygen concentration of the deep sea is directly related to carbon storage by the biological pump. The second paragraph of the main text provides a limited summary of some of the approaches that have been used to constrain past oxygen concentrations, but a much more thorough review of the evidence could be made here. Importantly, even though each proxy involves substantial assumptions and uncertainties, the variety of proxy methods that give consistent results showing lower oxygen concentrations throughout the deep ocean during glacial intervals affords great confidence to the findings. The conclusions of the paper would be much stronger if the authors emphasized this point about the consistency of diverse methods, and it could be done very concisely, just by listing in the main text the methods that indicate lower deep sea oxygen concentrations during glacial intervals (some references are provided below). The consistency among all of these records gives confidence in the conclusion that the climate-related cycle of atmospheric CO₂ reconstructed from ice cores is directly related to the amount of carbon stored in the deep ocean by the biological pump.

Response: We thank the reviewer for this assessment, and for the positive and helpful comments. As suggested, we have emphasized the consistency of results from diverse proxy methods (both published records and our new multi-proxy data) that gives confidence to our main conclusion about ocean carbon storage during late Pleistocene climate cycles (in the abstract, discussion, and conclusion sections). Following the other reviewers' request for more proxy data, we took additional samples from the studied core and measured redox-sensitive element concentrations (aU; Cd/Al, Mo/Al, U/Al) and benthic foraminiferal carbon isotope gradients, $\Delta\delta^{13}\text{C}$. Our new geochemical proxy data of oxygen reconstructions are consistent with the magnetofossil proxy record, which all reveal lower oxygen concentrations in the deep Indian Ocean during late Pleistocene glacials. We have considered the reviewer's suggestions and have incorporated the new results into the revised paper.

The most recent approach to reach these conclusions (that I am aware of) is to reinterpret the sub-surface manganese peaks, which have puzzled marine geochemists since at least the 1970s, as a further indicator of low oxygen concentrations in the deep ocean during glacial periods. Several recent papers have made this point, which has been summarized together with new data in a synthesis by Pavia et al [2021]. The

current paper by Chang et al. would be much stronger if its summary included this evidence.

Response: Thanks for this suggestion. We have added a brief summary of the recent synthesis of sub-surface manganese peak as an indicator of low oxygen concentrations by Pavia et al. [2021] (lines 48–49 of the revised paper). We have also added additional geochemical data for redox-sensitive metal concentrations (i.e., Cd/Al, Mo/Al) to support our interpretation of deep-sea oxygenation.

Figure 5 of the Chang et al paper is but one of many illustrations of the “nutrient deepening” hypothesis first posited by Boyle [1988a; b]. For example, illustrations similar to Fig 5 appear in Sigman and Boyle [2000] (their Fig 6) and in Bradtmiller et al [2010] (their Fig 6, which more than Sigman and Boyle 2000 is similar to Fig 5 in current paper). I recommend that these papers be acknowledged in the caption of Chang’s Figure 5.

Response: Thanks for pointing this out. The suggested papers have been cited and acknowledged in the Figure 5 caption.

Lastly, the results of this paper appear to someone who is not an expert in paleomagnetic methods to be very similar to the results of Korff et al [2016], both with respect to the conclusions about the impact of changes in bottom-water oxygen on the magnetic properties of sediments, and with respect to the time interval covered by the records. The Korff paper is cited in the SI (ref 16) but I recommend that it be worked into main text. Chang et al may wish to describe similarities or differences in methods, but most importantly I recommend that they describe the consistent conclusions reached by the two papers.

Response: As suggested, reference to Korff et al. [2016] has been moved to the main text. Descriptions of different methodologies and the consistency of conclusions reached by the two studies have been added (lines 47, 263–265 of the revised paper).

To reiterate, the paper is in fine shape and could be published in its present form, but the paper could have much greater impact if it provided a more complete review of relevant literature. Nearly 3 pages of text are currently devoted to magnetic results. If Chang et al are limited by the length of the paper, then perhaps some of the text concerning magnetic results could be moved to the SI to make space to accommodate the reference citations that I believe will make this a stronger paper.

Response: As suggested, we have moved some magnetic results text (e.g., ARM magnetic data, description of magnetic properties below 17.55 m due to magnetite dissolution, micromagnetic simulations) to the Supporting Information to make space to accommodate the suggested reference citations.

Key References:

Nutrient deepening hypothesis proposed – [Boyle, 1988a; b]

First paper to combine evidence from organic carbon flux proxy and sediment redox conditions to conclude that BWO must have been lower during the last glacial period – [Francois et al., 1997]

Clearest (in my opinion) explanation of the processes by which the ocean affects atmospheric CO₂, and graphical illustration of nutrient deepening – [Sigman and Boyle, 2000]

After a hiatus of more than a decade, the beginning of many studies that combined evidence from organic carbon flux proxy indicators and sediment redox conditions to conclude that BWO must have been lower during the last glacial period – [Jaccard et al., 2009] and [Bradt Miller et al., 2010]

Paper using magnetic properties of sediments to conclude that BWO was lower during glacial periods – [Korff et al., 2016]

Calibration and application of $\delta^{13}C$ of paired benthic foraminifera to conclude that BWO was lower under glacial conditions – [Hoogakker et al., 2015; Hoogakker et al., 2018]

Calibration and application of preservation of biogenic organic compounds to conclude that BWO was lower under glacial conditions – [Anderson et al., 2019]

Subsurface Mn peaks linked to lower BWO under glacial conditions – [Pavia et al., 2021]

Response: Thanks for this clear explanation of the relevant literature. These key references and associated descriptions have been incorporated into the introduction (lines 46–49 of the revised paper) and figure 5 caption.

Reviewer #2 (anonymous)

Chang and coauthors present a suite of magnetic measurements from marine sediment samples from the Eastern Indian Ocean, near the coast of western Australia, spanning the last 21 marine isotope stages (~850,000 years). From these magnetic datasets, they argue that the parameter δ_{BH} , or the relative proportion of biogenic hard (BH) to the total magnetofossil assemblage (BH plus biogenic soft (BS) magnetofossils), can be used to track changes in deoxygenation over glacial-interglacial cycles. They compare these data with several other datasets including transmission electron microscopy of magnetic extracts, micromagnetic modeling, x-ray fluorescence, atmospheric CO₂ concentrations, and bottom water oxygen calculations.

The manuscript is simply presented and well written. Overall, they do a nice job showing, interpreting, and defending their magnetic and electron microscopy results. For example, they put together convincing evidence from these datasets, supported by micromagnetic modeling, demonstrating that the relative increase in BH magnetofossils is due to a greater abundance of them, rather than other factors like

magnetofossil preservation that can affect these signatures. Unfortunately, I do not yet agree with their environmental interpretation that this increase in BH magnetofossils is directly linked to deoxygenation, which is one of the main takeaways of this manuscript. Before I can sign-off on this interpretation, I need to either be convinced that the increase in BH magnetofossils could not also be explained by other environmental changes (e.g., an increase in organic matter and or nutrient supply), and if they cannot rule out these interpretations then I would like to see caveats included. Although this is currently a weak spot in their manuscript, I still think this is an important contribution because they clearly demonstrate how BH magnetofossils are linked to environmental changes within and at the bottom of the water column during glaciation.

Response: We thank the reviewer for this assessment and for the thorough comments. We agree with the reviewer that ocean oxygenation and other environmental factors, e.g., organic carbon and nutrient supply, may contribute to the biomineralization of magnetite with different morphologies (lines 62, 70, 200, 207, 212, 314 of the revised paper). Following the reviewer's suggestions, we have made the following changes.

(1) We took additional samples from the studied core and measured further bottom-water oxygenation proxies, including the benthic foraminiferal carbon isotope gradient $\Delta\delta^{13}\text{C}$ and redox-sensitive trace metal concentrations (aU, Cd/Al, Mo/Al). These geochemical proxies for bottom-water oxygenation have been widely used. For example:

aU proxy: S. L. Jaccard, E. D. Galbraith, A. Martínez-García, R. F. Anderson, Covariation of deep Southern Ocean oxygenation and atmospheric CO_2 through the last ice age. Nature 530, 207–210 (2016).

Cd/Al, Mo/Al, U/Al proxies: J. Du, et al. Volcanic trigger of ocean deoxygenation during Cordilleran ice sheet retreat. Nature 611, 74–80 (2022).

$\Delta\delta^{13}\text{C}$ proxy: B. A. A. Hoogakker, et al., Glacial expansion of oxygen-depleted seawater in the eastern tropical Pacific. Nature 562, 410–413 (2018).

The results consistently indicate a pronounced bottom-water oxygenation decrease during glacials (revised Figure 4e, f). These new geochemical proxy data support our magnetofossil interpretation. Please also see our response to the other two reviewers on this point.

(2) We find more oxidized magnetofossils in glacial samples compared to interglacial samples from low-temperature magnetic measurements (Figure 2e, f), similar to previous studies (Chang et al., JGR 118, 6049–6065, 2013; Xue et al., JGR 127, e2022JB024714, 2022). Our data are consistent with the interpretation that oxygenation may be more important in controlling the magnetofossil morphology at this core site. We also added a description of the link between bottom water oxygenation and magnetite biomineralization, i.e. directly in the bottom waters or indirectly in sediments through oxygen diffusion.

(3) We do not have direct productivity and nutrient proxy data for the studied core. Data from nearby cores in the eastern Indian Ocean cores (e.g., core Fr10/95 GC17)

only indicate mild organic carbon supply changes during the last glacial-interglacial cycle (Murgese and De Deckker, 2007). Also, the core site is expected to have increased nutrient supply during interglacials (rather than during glacials) due to enhanced aeolian activity and monsoonal precipitation that brought more nutrients to the site (Stuut et al., Quat. Sci. Rev. 2014; Geophys. Res. Lett. 2019). Relevant discussions were added (lines 135–137, 231–241 of the revised paper). Also, even if organic matter supply makes a minor contribution to biomineralization of larger magnetofossils during glacials, our results are consistent with the interpretation of an overall enhanced carbon pool stored in the glacial Indian Ocean.

References cited here:

Chang, L. et al. Low-temperature magnetic properties of pelagic carbonates: Oxidation of biogenic magnetite and identification of magnetosome chains. J. Geophys. Res. Solid Earth, 118, 6049–6065, 2013.

Xue, P., Chang, L., Dickens, G. R., & Thomas, E. A depth-transect of ocean deoxygenation during the Paleocene-Eocene Thermal Maximum: Magnetofossils in sediment cores from the Southeast Atlantic. J. Geophys. Res. Solid Earth, 127, e2022JB024714. <https://doi.org/10.1029/2022JB024714>, 2022.

Murgese, S. D. & P. De Deckker. The late Quaternary evolution of water masses in the eastern Indian Ocean between Australia and Indonesia, based on benthic foraminifera faunal and carbon isotopes. Palaeogeogr. Palaeoclimatol. Palaeoecol., 247, 382–401, 2007.

Stuut, J.-B. W., Temmesfeld, F., & De Deckker, P. A 550 ka record of aeolian activity near North West Cape, Australia: Inferences from grain-size distributions and bulk chemistry of SE Indian Ocean deep-sea sediments. Quat. Sci. Rev. 83, 83–94, 2014.

Stuut, J.-B. W. et al. A 5.3-million-year history of monsoonal precipitation in northwestern Australia. Geophys. Res. Lett. 46, 6946–6954, 2019.

(4) We agree that other environmental factors, i.e., organic carbon and nutrient supply, may also affect magnetite biomineralization. We have added caveats about possible contributions from organic matter and nutrient supply to the Discussion.

In summary, we think that bottom-water oxygenation changes over glacial-interglacial cycles are the most important control of magnetofossil morphologies at the studied core site. This is supported by parallel redox-sensitive geochemical and benthic foraminiferal carbon isotope gradient oxygenation proxies. Discussion of possible contributions from organic carbon and nutrient supply has been added, as suggested by this reviewer.

Comments and requested revisions:

1. Looking at your data (FORCs with central ridge profiles, IRM unmixing, and TEM) it looks like the amount of biogenic soft (BS) magnetofossils remains constant or only decreases a small amount over these intervals, while the main thing really changing is

the relative proportion of biogenic hard (BH) magnetofossils. It is important that you explain this when you are presenting your data because, as written, I was expecting that the overall amount of BS magnetofossils would greatly decrease when the BH magnetofossils increased, but this is not the case. This is an important distinction because the environmental conditions that BS-producing MTB prefer must not disappear/go away when the BH-producing MTB become more abundant. Rather, the conditions that the BH-producing MTB prefer must develop and become persistent in addition to that of the BS-producing MTB. What do you think these conditions are? As an example, BS magnetofossils have been associated with oligotrophic environments and BH magnetofossil have been associated with seasonal nutrient/organic matter cycling (e.g., Egli, 2004). Therefore, I am not yet convinced of your interpretation here that the increase in BH is a direct indicator of “deoxygenation,” but rather a secondary indicator through an increase in organic matter or nutrient supply.

Response: The reviewer already pointed out the pronounced change in the relative proportion of BH magnetofossils (BH fraction δ_{BH} ; Figure 4e of the revised paper). We did not present down-core changes in absolute populations of BS and BH magnetofossils in the submitted paper. These data are now presented in Supplementary Figure S5. Our data, however, indicate a large BS magnetofossil concentration decrease (Supplementary Figure S5a) during glacial periods (rather than BS magnetofossils remaining constant), when the BH magnetofossils increased slightly during glacials (Supplementary Figure S5b). We suspect that the overall magnetofossil concentration decrease during glacials (BS + BH magnetofossils; Figure 4b; Supplementary Figure S5a,b) indicates a nutrient supply decrease for magnetite biogenesis. However, redox changes then mediate the relative proportion of BS/BH content at the studied site. Decreased glacial oxygenation during glacials is supported by oxygenation reconstructions from redox sensitive trace element concentration and benthic foraminiferal carbon isotope gradient proxies. We have clarified our statements on this and added additional discussion about nutrient /organic matter cycling in the revised results and discussion sections. Relevant statements for clarification were added in lines 135–137, 231–241 of the revised paper.

a. Please address this and either provide evidence against organic matter/nutrients in favor of your deoxygenation interpretation or acknowledge other potential causes for the increase in BH magnetofossils.

Response: This is a good suggestion – thanks for pointing it out. We acknowledge other potential causes for the BH magnetofossil increase and have added additional discussion of other environmental contributions, in addition to bottom-water oxygenation changes. Please see our response above.

b. Consider providing an explanation as to why the BS magnetofossils hardly show any change over the intervals.

Response: Please see our response above. Absolute BS magnetofossil content decreased

during glacials compared to interglacials. The overall magnetofossil concentration decrease may indicate decreased nutrient supply during glacials. We also observed a slight BS magnetofossil coercivity increase for glacial samples (Supplementary Figure S5c), as well as an overall size increase of isotropic and prismatic magnetofossils crystals (Figure 3 of the revised paper). These data are consistent with our interpreted decreased oxygenation during glacials.

c. Provide an explanation as to why we do not see a large increase in BH magnetofossils over every glacial event (Figure 4).

Response: Thanks for pointing this out. Reviewer 3 also raised this issue. As suggested, we now point out the different extent of δ_{BH} increase over different glacial intervals. From our data that span the last 21 marine isotope stages, an increased δ_{BH} is observed for most glacial-interglacial cycles; the glacial-interglacial contrast is unclear only for a couple of cycles (Supplementary Figure S7). We are not sure about the cause of this different degree of increase, but suspect that variable oxygenation and also possible contributions from organic carbon supply and others may produce variable BH magnetofossils increases over different glacial events. Statements have been added to the Result and Discussion sections (lines 135–137, 231–241 of the revised paper).

2. Related to and emphasizing comment #1, explore/consider some other interpretations for the increase in BH magnetofossils before you can land on a direct interpretation of deoxygenation from them, or be more clear about how you are exactly interpreting them. Correlation does not always mean causation. This is especially relevant over the intervals where you do not have a large increase in BH magnetofossils. Perhaps the environmental changes that lead to an increase in BH magnetofossils could have been stimulated by deoxygenation, or vice versa, but I do not think the BH magnetofossils can be directly interpreted to represent deoxygenation as presented. Your study area is located off the west coast of Australia, so it was not covered in any ice(?), and it also looks like three currents could have been affecting the study location: the East Gyral Current, the West Australian Current, and the Leeuwin Current. Please provide more information on these currents, what they are moving (cold/warm water? nutrients/nutrient-poor?) and how exactly they may affect the study location. I am unfamiliar with the study area, but given your results I would think that the increase in BH magnetofossils might indicate that the currents are more active and delivering organic matter and or nutrients. Could the increase in organic matter have created a more reducing environment, which would also have helped preserve magnetofossils and prevent oxidation (evidenced from your more pronounced Verwey transition)?

Response: Thanks for pointing this out. As suggested, we tried to make it clear that the magnetofossil proxy can be affected by organic carbon supply and that it provides an indirect oxygenation indicator. We have added descriptions of relevant ocean currents in the studied area. Discussion of possible enhanced organic matter and nutrients delivery is added, but ocean currents are suggested to have decreased during glacials in the studied area, which would bring less nutrients for magnetite biomineralization,

rather than a nutrient or organic carbon increase. Decreased ocean currents during glacials near the studied area are also supported by other paleoceanographic reconstructions (e.g., Murgese and De Deckker (2007) observed low glacial carbon isotope values of the benthic foraminifera species Cibicidoides wuellerstorfi in the eastern Indian Ocean, which was suggested to reflect reduced deep-water ventilation). Relevant statements were added (lines 231–241 of the revised paper).

Reference cited here: Murgese, S. D. & P. De Deckker, The late Quaternary evolution of water masses in the eastern Indian Ocean between Australia and Indonesia, based on benthic foraminifera faunal and carbon isotopes. Palaeogeogr. Palaeoclimatol. Palaeoecol., 247, 382–401, 2007.

Lines 57-58: Yes, redox conditions, but also other environmental changes like nutrients and organic matter cycling.

Response: These other environmental changes are now mentioned as suggested.

Lines 170-171: A good place to note that the abundance of BS magnetofossils does not change that much and emphasize that it's just a relative increase in the BH magnetofossils. Note that not all of the glacial intervals have large increases in BH magnetofossils (at least four have δ_{BH} values of less than 0.2).

Response: We now noted BS and BH abundance changes during glacials, with a relative BH magnetofossils increase. We also note that not all of the glacials had large BH magnetofossil increases.

Lines 172-176: To my knowledge, we do not have enough studies linking these different types of magnetofossils to make this claim about them being linked directly/only to oxygen levels (not including crystal maturity). I think most of the references you list here also say that organic matter supply could have been a main driver behind increases in BH magnetofossils. Also, it might be worth making it clearer that not all of your references are from glacial studies (e.g., #17, Chang et al., 2018). See comments above.

Response: We acknowledge that more studies are needed to link different magnetofossils to oxygenation. We have clarified that some studies are from other paleoclimate intervals (rather than glacial-interglacial cycles).

Lines 173-179: In general, this section is a bit confusing with how it is structured and linked to the references. Please re-write.

Response: We have re-written this section as suggested .

Lines 179-181: Note that elongated prismatic magnetofossils are more likely to behave as vortex particles, not single domain (e.g., see Wagner et al., 2021), which would contribute more-so to the backgrounds of FORC diagrams, rather than to the central ridge signatures.

Response: Thanks for pointing this out. The contribution of elongated prismatic magnetofossils to vortex behaviour and relevant reference of Wagner et al. (2021) is now mentioned in the Results (lines 82–84 of the revised paper).

Line 183: I don't think these references support what you are trying to say here. These references (#36 and 37) show that cuboctahedra particles, which are mostly equant and would contribute to the BS component, become more mature under more reducing conditions (Katzmann et al., 2013; Li & Pan, 2012).

Response: We have now made it clear that these are octahedral and cuboctahedral equant crystals. These references were cited here to demonstrate that generally increased oxygenation would suppress biogenic magnetite growth in magnetotactic bacteria. These issues have been clarified (lines 190–195 of the revised paper).

Lines 183-188: These references (#30 and 39) say that the BH magnetofossils were also linked to abundant organic matter (Chang et al., 2013; Yamazaki & Kawahata, 1998). Does an increase in organic matter always correspond to “deoxygenation”?

Response: More careful wording has been used to clarify. A possible contribution from organic matter is added (lines 199–200 of the revised paper).

Line 207: “This evidence” or “The evidence”

Response: Done.

Lines 207-209: How do you know there was not an increase in any type of organic matter or nutrient supply that leads to increased BH magnetofossils? What is the evidence against this?

Response: We removed the statement “rather than local signals caused by sedimentary organic carbon burial and respiration” and have added discussion about a possible organic matter or nutrient supply contribution to the increased BH magnetofossils.

Line 214: It's doesn't look like there's an exact match between panels (d) and (e) in Figure 4?

Response: We have added statistics of the correlation between our magnetofossil record and the Antarctic ice core record and the XRF record of the same core MD00-2361 (Supplementary Figure S7). Please also see our response to reviewer 3.

Lines 211-275: Please mention somewhere in this discussion section how the different currents affect the study area.

Response: As suggested, text has been added to the Discussion to show how the different currents affect the study area.

Line 236: “record indicates,” because you only have one record and you do not refer to

this in plural form when you mention it again later in the text (line 243)

Response: Thanks. The suggested text change has been made.

Line 239: Indicate where the two samples are from. Are these the red dots in Figure 4?

Response: We state in the Figure 3 caption that the stratigraphic position of the two studied TEM samples is indicated in Supplementary Figure S4.

Lines 253-256: This sentence is a little confusing. Consider re-writing.

Response: We have re-written this sentence as suggested (lines 286–288 of the revised paper)..

Line 262: “record”

Response: Done.

Lines 267-270: Here you link the deoxygenation with higher organic carbon fluxes. Why do you think the MTB are not directly linked to the increase in organic carbon, then?

Response: We agree that both bottom-water oxygenation and organic carbon flux can influence magnetite biomineralization. Texts describing this issue has been added. “Higher organic carbon fluxes” here refers to ocean water column processes. The text has been modified to clarify.

Lines 299 and 300: You mention that there is a difference in fluvial and eolian-derived detritus. What’s in each of them, why is this difference important for magnetotactic bacteria, and how could it affect the preservation of the magnetofossils?

Response: We currently do not know the difference in fluvial and eolian-derived detritus and their influence on magnetotactic bacteria. We suspect that this difference would not affect magnetofossil preservation in the main studied interval because magnetofossil dissolution only occurs below ~17 m in this core.

Line 342: Are these the red dots in Figure 4?

Response: These are not the red dots in Figure 4. The red dots in Figure 4 are benthic foraminiferal carbon isotope gradient oxygenation proxy data. We have marked the position of the two TEM samples in the sediment core (Supplementary Figure S4 of the revised paper).

Lines 345-346: Were the TEM images acquired randomly to prevent bias toward counting of BS or BH magnetofossils?

Response: Yes, the TEM images are randomly acquired, not specifically chosen. We have counted all magnetofossil grains in the TEM images, so that the results are not biased. We have added a statement about how we counted the magnetofossil grains in

the TEM images in the Methods (lines 279–281 of the revised paper).

Figures 1 and 2: Are these color-blind friendly palettes? Monochrome is always a good default option.

Response: Thanks for pointing this out. The color plots in Figure 1 were generated with the Ocean Data View program, which is useful for showing oxygen distributions. We have added O₂ concentration markers as contour lines so that the data are clear for color-blind readers. Text has been added to the Figure 1 caption to describe the isogram lines.

Figure 3: The difference in scale bars between the TEM images is a little misleading. Are all of the elongated particles in your TEM images magnetofossils or could some of them be something else like rutile? You might also consider labeling some of the morphologies to help your readers put a visual to the text when you mention these different shapes of magnetofossils (e.g., elongated prismatic, bullets, equant, etc.). Please double check you are consistent with your terminology about magnetofossil shapes. Also, check that your data match up. In the middle plots, the interglacial interval represented in panel (e) has particles listed with lengths >200 nm, but then the counts plot in panel (f) to the right doesn't go up that high. Expand the scale of the x-axis to include all the data points. Consider indicating where the additional BH magnetofossils fall (e.g., ~80-120 nm?) on the graphs to emphasize your point. Note that the red on green might be hard for color blind folks.

Response: (1) Some of the elongated particles in the TEM images are not magnetofossils, but carbonate materials. This is now clarified in the figure caption. (2) As suggested, we have labelled the different magnetofossil shapes (elongated prismatic, bullets, equant grains) in the TEM images. (3) We also checked through the terminology about magnetofossil shapes. (4) We use different colors in the grain size distribution histograms for the three magnetofossil types, so that their grain size distribution can be readily distinguished. (5) We use open red and solid green symbols in (b, e) for prismatic and equant magnetofossils.

Figure 4: Do the red dots correspond to the intervals used for magnetic extracts and TEM? Explain in the caption. Also, “biogenic” is misspelled on the y-axis of the bottom panel. Generally, it's a bit difficult to match up what you say in the text to this main figure because you mostly present data in either cm or m intervals (consider making this consistent in the text as well).

Response: (1) The red dots are benthic foraminiferal carbon isotope gradient proxy data, which is now explained in the figure caption. (2) Corrected to “biogenic”. (3) We have added depth profiles of the BH magnetofossil fraction δ_{BH} data in Supplementary Figure S4.

Figure 5: Where are the pink arrows? Are these meant to be the ones making the “U” shapes near the middle of the diagrams? This is your first mention of “polynyas,” so

please update your text to match this terminology, too.

Response: Thanks. (1) Text that mentions “pink arrows” has been removed from the figure caption. (2) “Polynyas” are now mentioned in the Discussion (lines 301–302 of the revised paper).

Reviewer #3 (anonymous)

The authors present a long magnetofossil record of bottom water oxygenation (BWO) in one sediment core from the eastern Indian Ocean. They interpret their record to reflect low BWO during glacial periods, which they believe suggests widespread carbon sequestration in the deep Indian Ocean (similar to what is thought for both the Pacific and Atlantic Oceans). I do not think that the current paper can be published in Nature Communications. First, I must point out that I am not an expert in the magnetofossil field. However, if I assume that the interpretation of their proxy is correct, I still have a problem with the interpretation of the data as presented for two reasons:

1) Only 2 samples were analyzed using the $\Delta\delta^{13}C$ proxy to corroborate their magnetic fossil interpretation. The two sample were analyzed at the Stage 10/11 boundary. The fact that their attempt to make this measurement at one other boundary (5/6 boundary) was unsuccessful is disconcerting. A glacial-interglacial BWO interpretation at one climate boundary using the $\Delta\delta^{13}C$ proxy is not a significant line of evidence. You would need more than a pair of data points to convince me, and not only from the same boundary (10/11) but from several boundaries. I believe their $\Delta\delta^{13}C$ data set needs to be expanded.

Response: We thank the reviewer for this assessment and for the helpful suggestions. As requested, we selected more samples across more glacial-interglacial cycles: MISs 3–5, MISs 12–9, MISs 19–21. We performed combined $\Delta\delta^{13}C$ on benthic foraminifera and redox-sensitive metal concentration measurements. New carbon isotope and geochemical data are all consistent with the magnetofossil proxy data. These new $\Delta\delta^{13}C$ and geochemical data are presented in Figure 4e, 4f of the revised paper. We also added statements about methods, data description and interpretation. Please see our response to the other two reviewers.

2) No statistical analysis is presented comparing the interpreted deoxygenation record in Figure 4e with the atmospheric CO_2 record in Figure 4d (this, arguably, is the crux of the manuscript), even though the authors say that the records are correlated. Perhaps there is a relationship between glacials and enhanced deoxygenation but it needs to be made explicit in a quantitative way. Also, if there is a relationship, it breaks down sometimes. For example, Stage 6 has a low biogenic BH fraction (should be high according to their interpretation) and Stage 7 has a high biogenic BH fraction (should be low according to their interpretation). Another example that goes against what the authors claim: in Stage 2 biogenic BH fractions decrease as atmospheric CO_2

decreases (the reverse should be the case according to the authors).

Response: We agree and have performed statistical analysis of correlations between different proxy data. We conducted linear regression analysis for biogenic BH fractions with the ice core atmospheric CO₂ record and XRF Ti/Ca ratio in core MD00-2361. Corresponding p-values of regression coefficients ($p < 0.01$) and the Spearman correlation coefficients ($R > 0.3$) indicate an apparent correlation between these variables (Supplementary Figure S7). We agree that this correlation can sometimes break down (this is now described in the revised paper). But considering these statistically significant and common correlations, we believe that the BH magnetofossil fraction δ_{BH} can effectively supports the relationship between glacial conditions and enhanced deoxygenation. We state this in the main text.

Some other comments:

What is the significance of the Ti/Ca record? I'm surprised that the strong correlation between the Ti/Ca and climate is not discussed. What is the cause of the relationship? I think the authors should do more with this data. Also, how do the geochemical relationships in the XRF data (Mn/Ti, Ti/Ca, S/Ti) relate to redox proxies in the sediment post-depositionally. If you can dissolve the biogenic magnetite at levels below 1770 m, why can't something similar be happening in the sediment intervals above 1770 m? Or, another way of putting this, why is diagenesis different below 1770 at this site?

Response: (1) The XRF elemental records (e.g., Ti/Ca ratio) for the studied core MD00-2361 have already been published by Stuut et al. (2014, 2019). The XRF data indicate a strong climate signal that was used by the above authors to trace aeolian activity and monsoonal precipitation in Northwestern Australia. We have added relevant discussions about these XRF data (lines 131–133 of the revised paper). (2) We interpret the main diagenetic change at ~1770 cm due to sulfate-reducing diagenesis (lines 144 of the revised paper) that dissolves iron oxides, including magnetofossils. Above ~1770 cm, the diagenesis is mild and magnetofossils are preserved. We have clarified this issue in the revision.

References cited here: J.-B. W. Stuut, F. Temmesfeld, P. De Deckker, A 550 ka record of aeolian activity near North West Cape, Australia: Inferences from grain-size distributions and bulk chemistry of SE Indian Ocean deep-sea sediments. Quat. Sci. Rev. 83, 83–94 (2014).

J.-B. W. Stuut et al., A 5.3-million-year history of monsoonal precipitation in northwestern Australia. Geophys. Res. Lett. 46, 6946–6954 (2019).

With respect to the bigger picture, other cores are implicated for having the same BH glacial enrichment, but none of the data from these other cores are shown in the figures. Why not present the data from the other cores to make your case stronger? For example, on lines 262-264 it's stated that the authors' record is broadly consistent with the

sediment proxy compilation and ocean modeling results. Why can't the compilation be made in a figure to show the broad consistency? Otherwise, the reader cannot assess the broad consistency claim that is made.

Response: Good suggestion. We have compiled the magnetofossil record of Yamazaki and Ikehara (2012) and plotted their results together with our data in the revised Figure 4e. These two BH magnetofossil fraction (δ_{BH}) records have consistent results with glacial enhancement of BH magnetofossils over a large area of the Indian Ocean.

REVIEWERS' COMMENTS

Reviewer #2 (Remarks to the Author):

Thank you to Chang and coauthors for providing a thorough rebuttal. I better understand the reasoning behind their interpretation of deoxygenation during glacial intervals. I specifically appreciate the explanation for why the authors think the overall increase in δBH is linked to deoxygenation at this site, rather than other environmental factors: evidence from previous studies shows a decrease in nutrient supply and organic matter during glacial periods coincident with increases in δBH . This justification was missing from the original manuscript.

In the revised manuscript, Chang and coauthors also add that the abundance of magnetofossils decreases over glacial cycles. They show that, although the abundance of magnetofossils increases during interglacial cycles, the relative proportion of BS increases while δBH decreases (note that this does not necessarily mean that the amount of BH decreased). The magnetofossil story here is interesting and disentangling it could be a separate paper. Additionally, the δBH proxy is a nice tool to look at relative trends in magnetofossil elongation in cores with consistent sedimentation/preservation.

My concern here is that the authors clarify their newly added statements. The way the text is written on lines 131-137 and 231-241 is a bit confusing to piece together. Although I remain skeptical that deoxygenation is the only control for the observed increase in δBH during glacial intervals, the combination of proxies for each of these intervals seems to be consistent with overall deoxygenation trends at this site. Please emphasize that, in this case, the combination of proxies allows for the interpretation of deoxygenation.

Reviewer #3 (Remarks to the Author):

After reading the responses to my review and the two other reviewers, I feel the authors have done an insightful revision of the manuscript. The authors took all concerns seriously and the manuscript has improved significantly as a result. I do recommend publication.

Response to Reviewer Comments

Reviewer #2 (anonymous)

Thank you to Chang and coauthors for providing a thorough rebuttal. I better understand the reasoning behind their interpretation of deoxygenation during glacial intervals. I specifically appreciate the explanation for why the authors think the overall increase in δ_{BH} is linked to deoxygenation at this site, rather than other environmental factors: evidence from previous studies shows a decrease in nutrient supply and organic matter during glacial periods coincident with increases in δ_{BH} . This justification was missing from the original manuscript.

In the revised manuscript, Chang and coauthors also add that the abundance of magnetofossils decreases over glacial cycles. They show that, although the abundance of magnetofossils increases during interglacial cycles, the relative proportion of BS increases while δ_{BH} decreases (note that this does not necessarily mean that the amount of BH decreased). The magnetofossil story here is interesting and disentangling it could be a separate paper. Additionally, the δ_{BH} proxy is a nice tool to look at relative trends in magnetofossil elongation in cores with consistent sedimentation/preservation.

My concern here is that the authors clarify their newly added statements. The way the text is written on lines 131-137 and 231-241 is a bit confusing to piece together. Although I remain skeptical that deoxygenation is the only control for the observed increase in δ_{BH} during glacial intervals, the combination of proxies for each of these intervals seems to be consistent with overall deoxygenation trends at this site. Please emphasize that, in this case, the combination of proxies allows for the interpretation of deoxygenation.

Response: We thank the reviewer again for this assessment, and for the helpful suggestions. As suggested, we have clarified the issue by modifying the statement to “Thus, we propose that, the combination of proxy results of increased magnetofossil δ_{BH} , decreased benthic foraminiferal $\Delta\delta^{13}C$, and increased redox-sensitive metal concentrations (Fig. 4e, f) during glacials compared to interglacials, are mainly controlled by decreased seawater oxygenation, while increased organic carbon and nutrient supply may only make a minor contribution at the studied core site” in lines 242–245 of the revised paper.

Reviewer #3 (anonymous)

After reading the responses to my review and the two other reviewers, I feel the authors have done an insightful revision of the manuscript. The authors took all concerns seriously and the manuscript has improved significantly as a result. I do recommend publication.

Response: We thank the reviewer again for the assessment.